# Van der Waals nanomesh electronics on arbitrary surfaces

You Meng [1,2,10], Xiaocui Li [1,3,10], Xiaolin Kang[1], Wanpeng Li[1], Wei Wang[1,2], Zhengxun Lai[1], Weijun Wang[1], Quan Quan[1], Xiuming Bu[1], SenPo Yip[4], Pengshan Xie[1], Dong Chen[1], Dengji Li[1], Fei Wang [5] ✉, Chi-Fung Yeung[6], Changyong Lan [7], Chuntai Liu [8], Lifan Shen[9], Yang Lu [3], Furong Chen[1], Chun-Yuen Wong [2,6] ✉ & Johnny C. Ho [1,2,4] ✉

Chemical bonds, including covalent and ionic bonds, endow semiconductors with stable electronic configurations but also impose constraints on their synthesis and lattice-mismatched heteroepitaxy. Here, the unique multi-scale van der Waals (vdWs) interactions are explored in one-dimensional tellurium (Te) systems to overcome these restrictions, enabled by the vdWs bonds between Te atomic chains and the spontaneous misfit relaxation at quasi-vdWs interfaces. Wafer-scale Te vdWs nanomeshes composed of self-welding Te nanowires are laterally vapor grown on arbitrary surfaces at a low temperature of 100 °C, bringing greater integration freedoms for enhanced device functionality and broad applicability. The prepared Te vdWs nanomeshes can be patterned at the microscale and exhibit high field-effect hole mobility of 145 cm$^2$/Vs, ultrafast photoresponse below 3 μs in paper-based infrared photodetectors, as well as controllable electronic structure in mixed-dimensional heterojunctions. All these device metrics of Te vdWs nanomesh electronics are promising to meet emerging technological demands.

In the past few decades, one-dimensional (1D) nanomaterials have been widely explored as the main driver for emerging electronics[1,2]. Particularly, nanowires (NWs) made of crystalline inorganic materials are able to meet all performance demands in terms of, but not limited to, carrier mobility, mechanical flexibility, energy efficiency, and optical transparency[3,4]. However, their scalability and integrability are still insufficient, especially when large-area and low-cost electronics are highly desired in present-day Internet of Things (IoT) applications[5]. Since then, many NW assembly strategies, including post-synthesis assemblies and in situ epitaxial growths, have been developed to tackle the issues of scalability and integrability by configuring large-scale crystalline NW parallel arrayed or random networked films.

Post-synthesis assemblies, such as contact printing and solution-phase deposition, are commonly used to assemble crystalline NWs into high-performance electronics and optoelectronics[6–8]. Taking advantage of post-synthesis assembly, several groups have developed circuit-level mechanically flexible applications, e.g., logic gates[9], ring oscillators[10], and artificial skins[11,12], using group III–V parallel-array NWs

[1]Department of Materials Science and Engineering, City University of Hong Kong, Kowloon 999077, Hong Kong SAR. [2]State Key Laboratory of Terahertz and Millimeter Waves, City University of Hong Kong, Kowloon 999077, Hong Kong SAR. [3]Department of Mechanical Engineering, City University of Hong Kong, Kowloon 999077, Hong Kong SAR. [4]Institute for Materials Chemistry and Engineering, Kyushu University, Fukuoka 816-8580, Japan. [5]State Key Laboratory of Luminescence and Applications, Changchun Institute of Optics, Fine Mechanics and Physics, Chinese Academy of Sciences, Changchun 130021, China. [6]Department of Chemistry, City University of Hong Kong, Kowloon 999077, Hong Kong SAR. [7]School of Optoelectronic Science and Engineering, University of Electronic Science and Technology of China, Chengdu 610054, P. R. China. [8]Key Laboratory of Advanced Materials Processing & Mold (Zhengzhou University), Ministry of Education, Zhengzhou 450002, P.R. China. [9]College of Microelectronics and Key Laboratory of Optoelectronics Technology, Faculty of Information Technology, Beijing University of Technology, Beijing 100124, P.R. China. [10]These authors contributed equally: You Meng, Xiaocui Li. ✉e-mail: wangf@ciomp.ac.cn; acywong@cityu.edu.hk; johnnyho@cityu.edu.hk

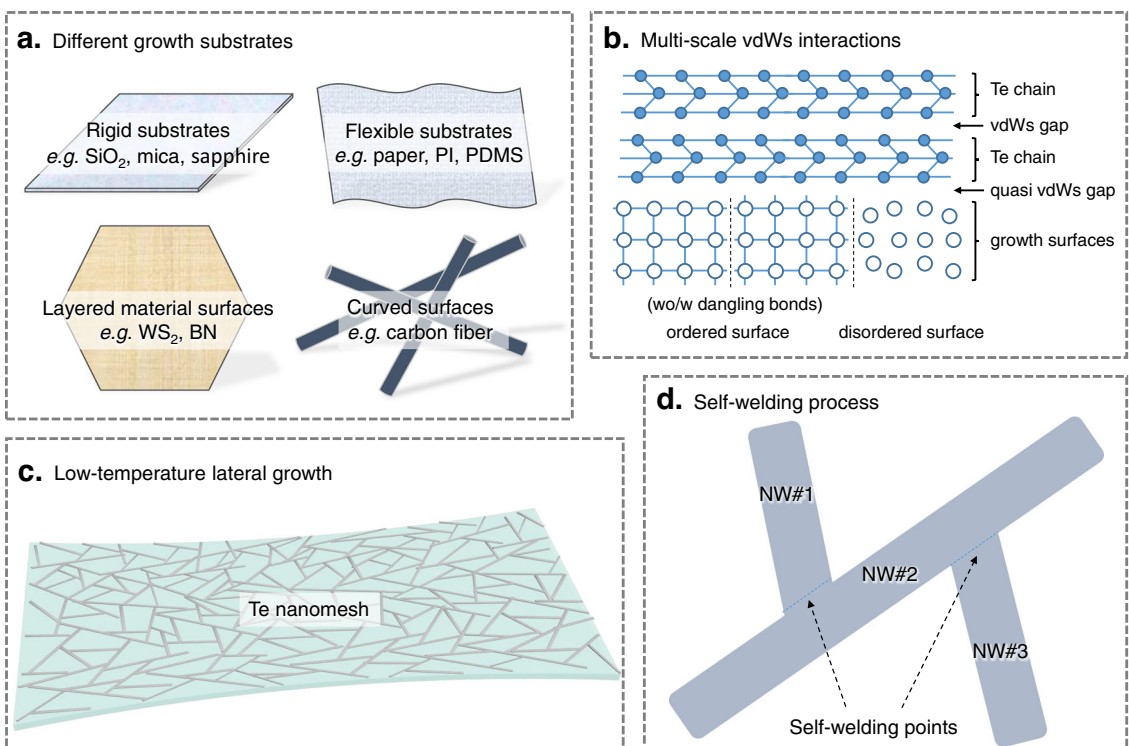

**Fig. 1 | Schematic illustrations of the low-temperature Te nanomesh growth on arbitrary surfaces. a** Growth substrates used in this work with different forms, shapes, and sizes. **b** Multi-scale van der Waals (vdWs) interactions of Te nanomeshes. **c** Low-temperature lateral growth of Te nanomeshes on heat-intolerant substrates. **d** The self-welding process between Te nanowires (NWs).

and carbon nanotube (CNT) thin films as active components. It is noted that although these assembly methods are generally applicable to different 1D nanomaterials, the cost-intensive and slow process natures would hinder the practical utilization of post-synthesis assembly processes[5]. Alternatively, to bypass the complicated two-step assembly process, in situ epitaxial growths are intensely studied to construct 1D materials in a bottom-up manner and explore their functional devices[3,13]. In some cases, for instance, gallium nitride (GaN) NWs with controlled crystallographic orientations are grown on specific crystal planes of sapphire ($\alpha$-Al$_2$O$_3$)[13]; however, the high growth temperature, stringent substrate requirement of atomically flat and faceted surfaces, and elusive heteroepitaxial relationships are always involved, which add apparent complexities to the standard material processing.

Under these considerations, alternative materials and growth strategies are desired to address the current challenges of 1D nanomaterials. Recently, elemental tellurium (Te) microplates[14,15], thin films[16,17], and 1D chains[4,18] are demonstrated good electrical device performance with good environmental stability and high hole mobility, mainly arising from their narrow band gap (0.35 eV) and high current-carrying ability. As a kind of "DNA-like" inorganic molecules, 1D Te atomic chains are packed in a hexagonal array by weak forces, called van der Waals (vdWs) interactions, to form a crystal, while in each atomic chain, the covalently bonded Te atoms are helically sequenced along the *c*-axis[14]. The vdWs interactions, together with the highly anisotropic crystalline structure of Te crystal, would potentially dictate the 1D nanostructured growth. Such a true 1D system, consisting of vdWs bonded atomic chains, provides us opportunities to overcome those severe constraints on fabricating traditional 1D materials[18].

Here, aiming to bypass the constraints on semiconducting material synthesis and lattice-mismatched hetero-integration, we report a robust vapor-phase growth method to synthesize wafer-scale Te vdWs nanomeshes at only 100 °C. Enabled by the vdWs bonds between Te atomic chains and the spontaneous misfit relaxation at quasi-vdWs

interfaces, Te vdWs nanomeshes could be laterally vapor grown on arbitrary surfaces, including rigid substrates, flexible substrates, layered material surfaces, and curved surfaces. The multi-functionality of self-welded Te vdWs nanomeshes is also demonstrated, such as micrometer-level patterning capacity, high field-effect hole mobility, fast photoresponse in the infrared region, and controllable electronic structure of heterojunctions. All the obtained device metrics are on par with those of state-of-the-art devices.

## Results
### Low temperature and general growth
To initiate the growth process, Te source powders are vaporized and carried to the growth substrates, heating at 100 °C with a flow of argon gas (details provided in the "Methods" section). No metal catalyst is used to seed the NW growth in this work. The growth pressure is atmospheric pressure during the entire growth process, avoiding the capital-intensive equipment usually employed in conventional synthesis techniques. The atmospheric pressure is also beneficial to suppress the evaporation of source materials and yield an efficient NW nucleation/growth with uniform morphology. As illustrated in the schematic diagrams in Fig. 1, the low-temperature Te nanomesh synthesis offers many attractions in terms of material, process, and device. Highlights of this work are listed below:

i.  The self-assembled Te nanomesh could be grown on various types of surfaces, including rigid substrates, flexible substrates, layered vdWs material surfaces, and three-dimensional curved surfaces (Fig. 1a). There is no concern about the compatibility between the target device substrate and the NW growth process; thereby, devices can be made on diverse technologically functional surfaces in a scalable and low-cost manner that cannot be achieved by other means.

ii. The Te nanomesh synthesis presented in this work is simply based on the multi-scale vdWs interactions, i.e., the macroscale quasi-vdWs interactions with spontaneous misfit relaxation to

substrates[19] and the nanoscale vdWs interactions among Te chains[4,18] (Fig. 1b and Supplementary Fig. 1). Unlike the growth mechanisms proposed for metal-catalyzed or epitaxial growth that mainly rely on the complicated liquid–solid interface and epitaxial relationship, our multi-scale vdWs interactions-dominated growth is spontaneous, efficient, and multi-substrate compatible.

iii. Because all the vdWs interactions explored here are parallel to the growth substrates, the Te NWs would laterally grow on the substrates and then assemble into NW networks. This laterally grown nanomesh is beneficial to the subsequent device integrations, as most present-day devices are fundamentally planar in architectures[5]. Moreover, our low-temperature nanomesh fabrication approach would not involve any chemical reaction and associated damage on typical substrate materials (Fig. 1c). Therefore, directly growing Te nanomeshes on heat-intolerant flexible or stretchable substrates, such as polyimide (PI) and polydimethylsiloxane (PDMS), is now accessible.

iv. A self-welding process is found in our Te nanomesh growth process, which is essential for promoting electrical device performance and mechanical robustness (Fig. 1d). Compared to the NW networks formed by solution-phase deposition schemes that always give the weak physical inter-NW connection[20], our self-welded nanomeshes with well-connected network morphology are expected to exhibit reduced inter-NW junction resistance and improved mechanical robustness, making them a kind of potentially reliable active device component for practical applications.

## Geometric control and self-welding process

Taking sapphire substrate as an example, nucleation, lateral growth, and self-welding steps of Te NWs occur sequentially throughout the entire growth process (Fig. 2a, b). To start with, Te molecules randomly diffuse on the growth surface and reach an active surface site (steps, defects, impurities, etc.) that is energetically favorable for nucleation[21]. As the densities of active surface sites depend on the substrate type and quality, we observed enhanced nucleation densities on roughened surfaces (Fig. 2c and Supplementary Fig. 2) because of their increased surface defects. In specific, the nucleation densities on PDMS and photoresist film are found to be $4.5 \pm 0.2\,\mu m^{-2}$ and $3.5 \pm 0.2\,\mu m^{-2}$, respectively, significantly higher than those on some surface-clean substrates (e.g., $0.12 \pm 0.09\,\mu m^{-2}$ of $SiO_2$ and $0.45 \pm 0.15\,\mu m^{-2}$ of sapphire). Although with distinctly different substrate features, Te nanomeshes on both rigid and flexible substrates show basically similar growth behaviors (Supplementary Fig. 2). Besides, because of the pronounced atomic-chain crystal structure, the Te atoms are more strongly coupled with each other in an atomic chain than with the growth substrate. Following the Volmer–Weber nucleation model[22,23], the nanoclusters are nucleated and grown directly on the substrate surface, regardless of the surface features, giving rise to the general growth of Te nanomeshes on diverse substrates.

To substantiate the correlation between nucleation density and substrate quality, we performed contact angle measurements on various vacant substrates. The contact angle could provide quantitative data about the molecular differences in surface energy (Fig. 2c). In general, the active surface sites would create a low-energy surface, lowering the energy barrier for nucleation. Indeed, we observed enhanced nucleation densities on surfaces with low surface energies (Fig. 2c). This finding allows us to modify the NW density of Te nanomeshes via surface energy engineering. The control experiments are demonstrated to endorse the proposed correlation (Fig. 2d). For instance, to increase the NW density on certain substrates, HF-treated $SiO_2$ (changing terminated groups), 1000 °C-annealed sapphire (creating steps), Ar plasma-bombarded PI (increasing roughness) were conducted to lower the corresponding surface energies. On the contrary,

enhancing the surface energy of $SiO_2$ by removing surface carbon contaminants via $O_2$ plasma treatment would lead to decreasing NW density. These results are in good agreement with the proposed surface energy-determined nucleation density correlation and also specify some defects/impurities that could directly trigger the nucleation and growth of Te NWs, including surface groups, steps, and contaminants.

With a continuous supply of Te adatoms, the lengths of Te NWs are increasing with the growth time, while at the same time, the NW diameter is relatively constant (~150 nm) (Fig. 2a, b). This result reveals that the Te incorporation in the radial direction and axial direction of Te NWs is significantly different. From transmission electron microscopy (TEM) images and selected area electron diffraction (SAED) patterns (Supplementary Fig. 3), single crystallinity and growth orientation of $\langle 0001 \rangle$ for the Te NWs are distinguished. The favored $\langle 0001 \rangle$ growth of the Te NW is mainly owing to the highly anisotropic crystalline structure of Te, agreeing well with the reported works[4,21]. The covalently bonded Te atoms along the $c$-axis have higher binding energy (0.68 eV) than those along the $\langle 1\bar{2}10 \rangle$ (0.22 eV) and $\langle 10\bar{1}0 \rangle$ (0.05 eV) directions[21]. Therefore, in a continuous adsorption and desorption process, driven by the stronger bond in $\langle 0001 \rangle$ directions, the growth along the $c$-axis (i.e., the NW axial direction) is kinetically favored, while the growth along the radial direction almost stops with a constant diameter of 150 nm. Changing the local energetic landscape with growth temperatures would directly affect the NW morphology, including diameter, length, and density. As shown in Supplementary Fig. 4, with decreased growth temperatures from 200 to 50 °C, the Te NWs possess increased density but decreased length and decreased diameter from ~3 μm to ~20 nm. For benchmarking with the state-of-the-art ultrathin inorganic chains, future NW scaling with significantly reduced NW diameter could be realized by growth optimization and subtractive manufacturing (e.g., etching and exfoliation)[18].

By increasing the growth duration time to 2 h or above, Te NWs extend to connect with each other, and the self-welding processes appear at the inter-NW junctions (Fig. 2e). As depicted in the TEM image in Supplementary Fig. 5, the atomic-sharp boundary formed between two neighboring Te NWs confirms the self-welding process of Te nanomesh. The welding angles among NWs are in the random values, as there have no coherence and lattice epitaxy relationship between NW and substrate. From the TEM study on the welding point of two NWs, the welding angle is consistent with their contact angle determined by their growth orientations. As for the self-welding mechanism, this could be understood as two microstructures are closely in contact with each other; a spontaneous vapor condensation occurs, followed by a welding point formation at the contact area. To better understand the self-welding process in this work, schematic illustration (Fig. 2e), SEM images (Fig. 2f), and TEM images (Supplementary Fig. 6) were displayed with different growth duration times. Compared to the 2- to 3-h-grown nanomeshes, the enlarging soldering points and the thicker NW body are clearly observed with prolonging growth durations above 4 h due to the additional preferred vapor condensation, showing the dramatically improved welding status and thus better electrical properties (Supplementary Fig. 7). This is consistent with the reported experiments[24] and simulations[25] that the microstructures with small curvatures or nanoscale gaps would be the preferred sites for vapor condensation. The vapor condensation-induced welding process has been proposed to enhance the conductivity as well as mechanical strength of solution-processed metal NW networks[26].

The compositional purity and hexagonal structure are also carefully evaluated by X-ray diffraction (XRD), energy-dispersive X-ray spectroscopy (EDS), and Raman study in detail (Supplementary Fig. 8). All these results can evidently indicate the impressive material properties of Te nanomeshes, together with the well-defined NW morphology and uniform nanomesh assembly. Importantly, as shown in Supplementary Fig. 9, the XRD patterns of Te nanomeshes illustrate the unaltered hexagonal-phase diffraction peaks, both in peak

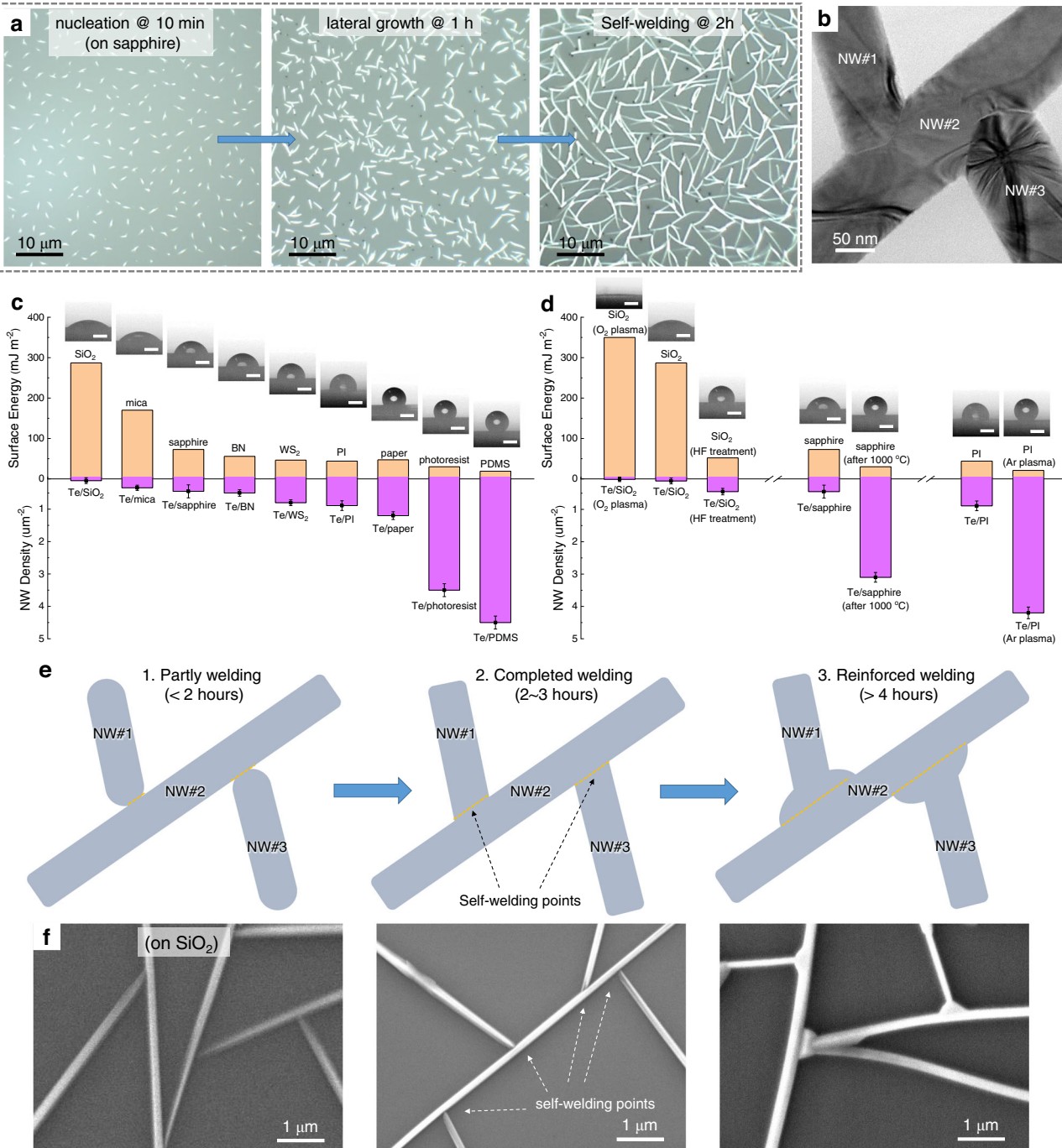

**Fig. 2 | Te nanomesh growth. a** Optical images of Te nanomeshes grown on sapphire with different growth durations, showing nucleation, lateral growth, and self-welding steps. **b** Transmission electron microscopy (TEM) images of Te nanomeshes with distinct self-welding regions. **c** Surface energies of different substrates used in this work and corresponding NW densities of Te nanomeshes after growth. The corresponding references could be found in Supplementary Table 1. The inset shows photographs of water droplets on different substrates.

Scale bar represents 2 mm. **d** Surface energies of different substrates before/after surface treatments and the corresponding NW densities of Te nanomeshes after growth. The inset shows photographs of water droplets on different substrates. Scale bar represents 2 mm. **e** Schematic illustration and **f** scanning electron microscopy (SEM) images of the self-welding process of Te nanomeshes with different growth duration times.

intensities and peak positions, even remaining unchanged for storage in ambient conditions for 300 days. While the Te nanomesh transistors are fabricated, the device performances show no discernible degradation in output current or mobility after 50-days storage in ambient, even without device encapsulation (Supplementary Fig. 10). These observations highlight the superior environmental stability of Te nanomeshes against environmental factors (e.g., moisture and atmospheric oxygen), which is highly desired for practical applications. The

naturally terminated dangling-bond-free surfaces of vdWs nanomeshes play a key role in blocking the surface-induced material degradation commonly observed in typical nanoscale semiconductors[15].

## Multi-scale vdWs integration

We postulate that the multi-scale vdWs interactions, i.e., the macro-scale quasi-vdWs interactions to substrates[19] and the nanoscale vdWs

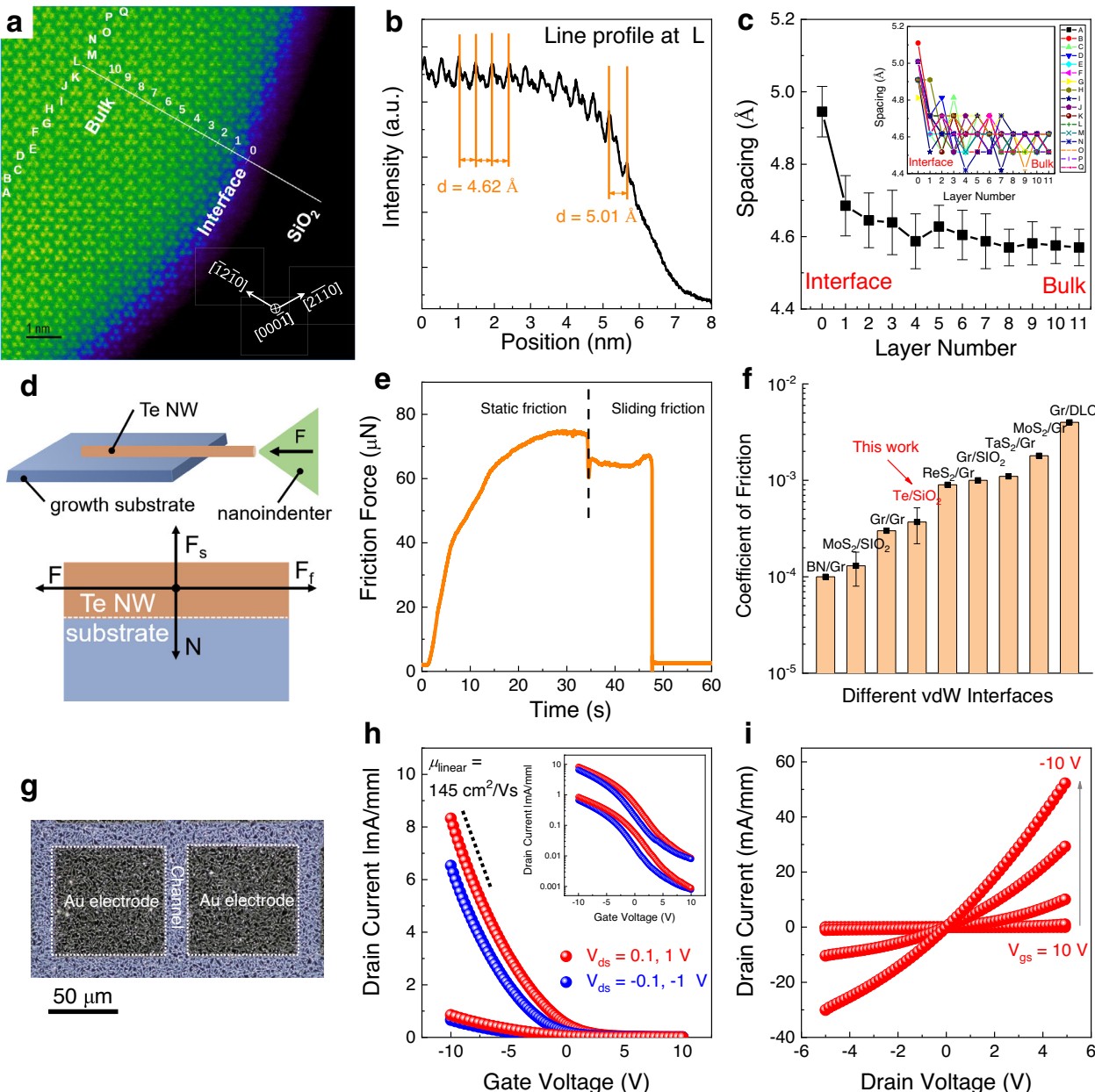

**Fig. 3 | Multi-scale vdWs integration. a** Cross-sectional high-angle annular dark field scanning transmission electron microscopy (HAADF-STEM) image of Te NW/SiO₂. **b** Line intensity profile located at L position in the HAADF-STEM image. **c** The average plane spacing of ($\bar{1}2\bar{1}0$) as a function of layer number ranging from interface to bulk. The measurement was reproduced on 17 samples and the error bar represents the standard deviation. Inset of (**c**) shows the measurement results of 17 samples. **d** Schematic illustration of the friction property test. F, Fₛ, F_f, and N represent the load force, supporting force, friction force, and cohesive force,

respectively. **e** The corresponding friction force measured in the friction property test. The vertical dashed line indicates the boundary between the static friction process and the sliding friction process. **f** Comparison of the coefficient of friction of various vdWs interfaces. The corresponding references could be found in Supplementary Table 2. **g** Optical image, **h** transfer, and **i** output characteristic curves of the p-type Te nanomesh thin-film transistors (TFTs) fabricated on SiO₂. Inset of (**h**) shows the transfer curve using logarithm y-coordinate.

interactions among Te chains[4], dominate the growth process of Te nanomeshes. Unlike ionic or covalent bonds, the vdWs forces (or vdWs interactions) exist between the neighboring atoms, molecules, and surfaces ubiquitously. In individual atomic-chain Te NWs, the nanoscale vdWs interactions between adjacent chains would promote the anisotropic growth of NWs along the chain direction. Moreover, since most growth substrates used in this work are three-dimensional materials with reactive dangling bonds while the Te NW surface is free of dangling bonds, the binding mechanism between such dissimilar materials is referred to quasi-vdWs interactions[19,27]. To shed light on the atomic structure of such quasi-vdWs interface, high-angle annular

dark field scanning transmission electron microscopy (HAADF-STEM) was conducted on a Te NW/SiO₂ interface in cross-sectional view. As shown in the HAADF-STEM image (Fig. 3a), spiral Te atomic chains are oriented along the c-axis of the hexagonal array, together with the distinct interface to SiO₂. The intensity profile was analyzed to show the ($\bar{1}2\bar{1}0$) plane spacing in Fig. 3b. Interestingly, the first plane spacing at the interface is 5.01 Å, being significantly larger than the normal bulk plane spacing of 4.62 Å.

In general, strain can relax by plastic deformations or elastic dislocations, in a short range or a long range[28,29]. In this work, the strain induced by the interfacial lattice mismatch is substantially eliminated

within one nm (i.e., two vdWs blocks) near the interface (Fig. 3c) via the re-configuration of Te chain spacings, indicating an abrupt interfacial relaxation of the misfit strain, i.e., short-range plastic deformations (Supplementary Fig. 11)[28]. This is in contrast to the conventional heteroepitaxy in lattice-mismatched systems, where the misfit strain relaxes gradually via the introduction of bulk dislocations to release the accumulated elastic energy, i.e., long-range elastic dislocations[30]. Considering the strength of weak quasi-vdWs interaction (0.1–10 kJ/mol) is about two or three orders of magnitude less than that of chemical bonds (100–1000 kJ/mol)[31], the immediate strain relaxations happened at the quasi-vdWs interface efficiently through short-range plastic deformations, other than the formation of long-range elastic dislocations within NW bulk. The spontaneous near-interface relaxations induced by the weak quasi-vdWs interaction may enable the general growth of Te nanomeshes on various substrates.

The superlubricity phenomenon has been widely found in vdWs interfaces, featuring the ultra-low coefficient of friction (COF) below $10^{-3}$, enabled by their weak vdWs interactions and natural lattice mismatch[32]. Here, the frictional properties between Te NW and $SiO_2$ growth substrate were studied (Fig. 3d and Supplementary Fig. 12), by which direct experimental evidence for the vdWs interaction-induced superlubricity was found. The friction property test on Te/$SiO_2$ can be achieved by the lateral movements of the diamond nanoindenter to push the Te NW, where a displacement control mode with a constant loading velocity of 20 nm/s is used. Evidently, both the static friction process and the sliding friction process are observed in Fig. 3e, where the friction force ($F_f$) is up to 75 μN. The cohesive force (N) is mainly contributed by the vdWs interface energy, which can be calculated by $N = \sigma S/A$[33], where σ is interfacial adhesion energy with the typical values of 0.1–0.4 J cm$^{-2}$[34], S is the contact area of 200 μm$^2$, and A is the interaction distance of 0.2 Å between Te NW and substrate. As a result, the COF = $F_f$/N is estimated to be $3.7 \pm 1.5 \times 10^{-4}$, being comparable to most vdWs materials and their heterostructures (Fig. 3f). On the macroscopic scale, the typical peel-off process of Te nanomeshes from different growth substrates to target substrates were demonstrated (Supplementary Fig. 13), also highlighting the macroscale quasi-vdWs interactions of Te nanomeshes/substrates. These results directly prove the low-energy quasi-vdWs interaction between Te nanomeshes and growth substrates.

Overall, the multi-scale vdWs interactions, from the macroscale self-assembly to the nanoscale chain conformation, are fundamentally different from the growth mechanisms proposed for metal-catalyzed growth or epitaxial growth. The quasi-vdWs interactions with spontaneous misfit relaxation are also different from the conventional van der Waals epitaxy (vdWE) growth, in which the chemically inert surface needs to be dangling bonds free or be terminated regularly[31]. In contrast to that, different surfaces, either with or without dangling bonds, are capable of Te nanomesh growth in our work. Equally importantly, as both macroscale and nanoscale vdWs interactions are parallel to the growth substrates, the Te NWs would laterally grow on the substrates and further facilitate the subsequent self-welded nanomesh formation for extended applications. With thermodynamic considerations, the energy minimizations also play a key role in determining the lateral Te NWs growth on substrates (Supplementary Fig. 14)[35]. Here, because the packed Te molecular chains are in the plane of the growth substrates, the (0003) diffraction peak is absent in the XRD patterns (Supplementary Figs. 8 and 9).

## P-type transport properties

To evaluate the charge transport characteristics of Te nanomeshes, thin-film transistors (TFTs) based on Te nanomeshes are fabricated on the $SiO_2$/p$^{++}$ Si (50-nm-thick thermal oxide) substrates (Fig. 3g). To complete the bottom-gate top-contact device structure, the 5/80-nm-thick Cr/Au electrodes defined by shadow masking are adopted as the source/drain electrodes. From the transfer and output characteristic curves of Te nanomesh TFTs, the conductance of the nanomesh channel is effectively modulated by the gate bias voltage and channel length (Fig. 3h and Supplementary Fig. 15). Because of the gate screening effect, typical asymmetric transfer curves when applying positive and negative source-drain ($V_{ds}$) voltage are found in our p-type Te nanomesh TFTs (Fig. 3i). Due to the same reason, the output curves show the obvious saturation behavior at large negative $V_{ds}$, while upon positive $V_{ds}$, the device does not show the sign of saturation.

When operating in the linear regime, the device demonstrates good p-type transistor performance with a peak field-effect hole mobility of 145 cm$^2$/Vs (detailed mobility extraction shown in the "Methods" section). This mobility is comparable to or even better than conventional p-type semiconducting thin-film devices, including metal oxides, metal halides, perovskite halides, organic materials, and CNT thin films, as summarized in Supplementary Table 3. From the statistical analysis of 20 devices (4 × 5 arrays) on the same wafer presented in Supplementary Fig. 16, the on-current of $7.5 \pm 1.2$ mA/mm, the threshold voltage of $1.8 \pm 0.3$ V, and the peak hole mobility of $145 \pm 15$ cm$^2$/Vs are obtained, further proving the good uniformity of nanomesh TFT performance. The high hole mobility and high current-carrying capacity are mainly attributed to the single-crystalline nature of individual Te NW and the good electrical connections between self-welded NWs.

Based on a rationally designed vdWs nanomesh deposition method, the morphologies of Te nanomesh could be controlled in this work, which would in turn boost the device performances, including mobility and output current, to a promising level. To better show the growth control and corresponding device performance optimization, the related experiments were carried out as a function of different substrate selections, deposition temperatures (Supplementary Fig. 17), and growth durations (Supplementary Fig. 7). With the tunable diameter, length, density, and welding status of Te nanomeshes, we could obtain a powerful platform to obtain an on/off current ratio of ~10$^3$ and a hole mobility up to 195 cm$^2$/Vs. The obtained hole mobility value outperforms all the scalable Te-based materials, like the evaporated Te thin films of 35 cm$^2$/Vs[16] and Te$_x$Se$_y$ thin films of ~40 cm$^2$/Vs[17], but lower than some single Te nanostructure, like solution-grown 2D Te layers of 700 cm$^2$/Vs[14] and 1D Te NW encapsulated in nanotube of 600 cm$^2$/Vs[4]. This result is reasonable when considering wafer-scale coverage and inter-NW connections among nanomeshes. It is worth mentioning that the theoretical mobilities of Te materials in literature are up to ~10$^5$ cm$^2$/Vs[36], arising from the narrow bandgap and low effective mass. Thus, there is still ample spacing to boost further the mobility of Te nanomeshes with various strategies, such as surface doping[37], composition design[17], and dielectric engineering[38].

## Multi-substrate compatible growth

In order to show the universality of nanomesh methodology, the direct forming of self-welded Te nanomeshes on various types of surfaces is demonstrated, including rigid substrates (sapphire, Fig. 2a; $SiO_2$, Fig. 2f; and mica, Fig. 4a), flexible substrates (PI, Fig. 4b; paper, Fig. 4c; and PDMS, Fig. 4d), layered 2D material surfaces (tungsten disulfide, Fig. 4g and boron nitride, Fig. 4h), and three-dimensional curved surfaces (carbon fibers, Fig. 4i). These implemented substrates are across different categories regarding whether they are ordered or disordered, rigid or flexible, bulk or layered surface, flat or curved, etc. Specifically, the direct Te nanomesh growth on heat-intolerant substrates is successfully realized with a temperature as low as 100 °C, completely unattainable by other vapor growth methods of 1D nanomaterials. Moreover, the unique capability of multi-substrate compatible growth is also used to grow nanomeshes on carbon fibers, a kind of curved substrate (Fig. 4i and Supplementary Fig. 18). This multi-substrate compatible growth is difficult to be achieved by the post-synthesis assembly of NW networks relying on mechanical assembly or solution-phase deposition schemes[5].

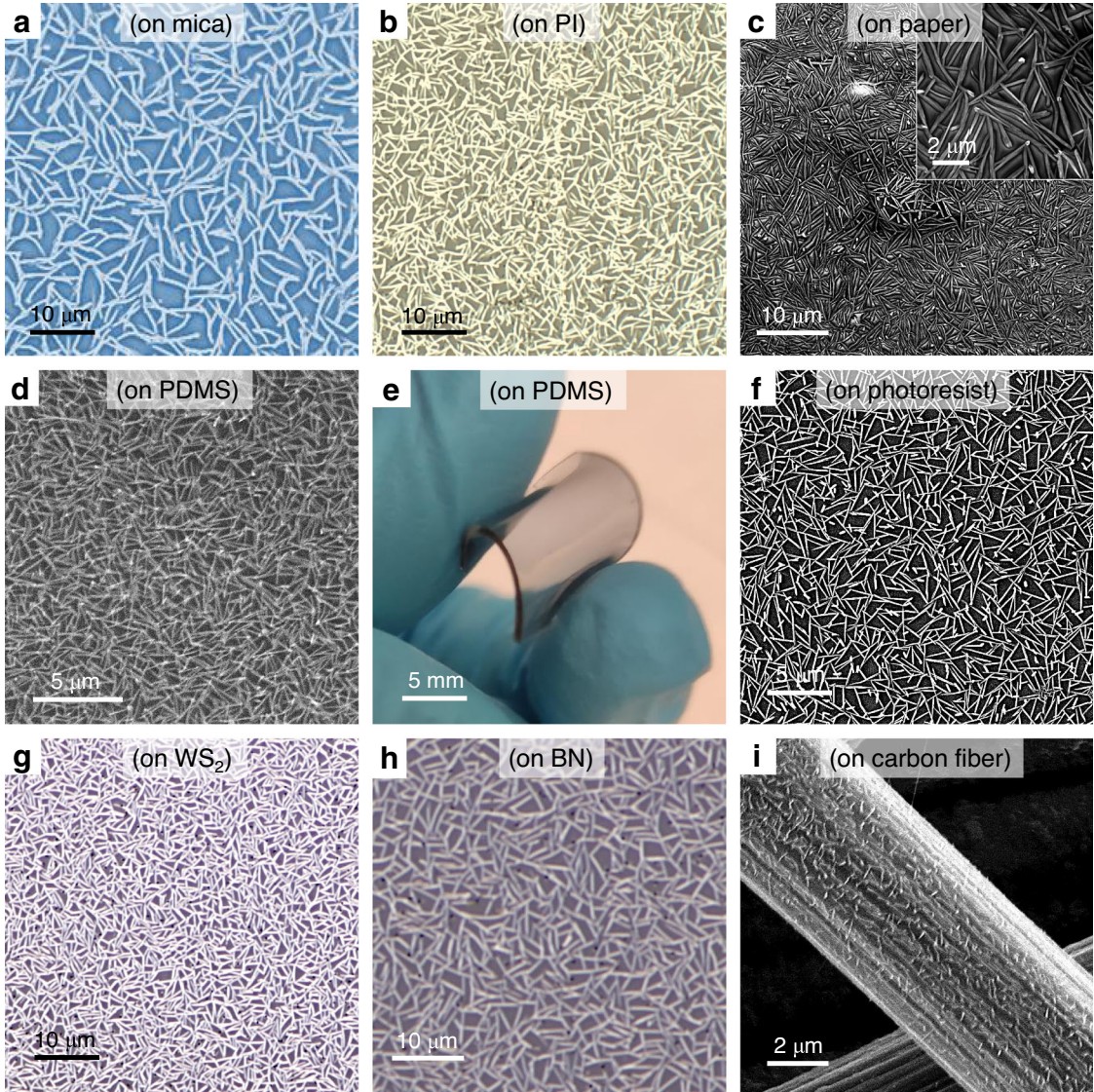

**Fig. 4 | Multi-substrate compatible growth.** Optical and SEM images of Te nanomesh grown on different surfaces, including **a** mica (optical), **b** PI (optical), **c** paper (SEM), **d** PDMS (SEM), **e** PDMS (photograph), **f** photoresist (SEM), **g** WS$_2$ (optical), **h** BN (optical), and **i** carbon fiber (SEM). Inset of (**c**) shows the enlarged SEM image of Te nanomesh grown on a paper substrate.

The relationship between crystallinity and electrical properties of the obtained Te nanomesh on different substrates were also studied, where their crystallinities and conductivities of Te nanomesh witness a decreasing trend from SiO$_2$, mica, sapphire, polyimide to paper (Supplementary Fig. 19). Afterward, the synthesis scalability is also investigated by employing a vapor transport system equipped with a 2-inch quartz tube. As a result, wafer-scale Te nanomesh is reliably yielded on different kinds of substrates (Supplementary Figs. 20 and 21). Anyway, despite the significantly different surface forms, shapes, and sizes, all these substrates are fully covered by the well-connected nanomesh after growth, further highlighting the versatility of vdWs nanomesh for different utilization.

**High-resolution nanomesh patterning**

Microscale patterning is a fundamental step in the implementation of nanomaterials for practical devices. Traditional macroscale integration of crystalline NWs via conventional patterning methods usually suffers from low resolution and poorly defined pattern edges. In this work, we have successfully demonstrated a nanomesh patterning process with a micrometer-level resolution by combining the multi-substrate compatible nanomesh growth and photolithography process (Fig. 5a).

First, patterned photoresist films were generated using a standard exposing/removing process (details shown in the "Methods" section). Taking advantage of the multi-substrate compatible growth, Te nanomesh were grown on both the pre-patterned photoresist film and the photoresist-uncovered SiO$_2$ substrates simultaneously, as shown in the SEM image in Supplementary Fig. 22. After the nanomesh growth, the patterned photoresist film and the nanomesh on it were removed at the same time through a lift-off process. Finally, the nanomesh grown on the uncovered substrates remained to project the microscale images onto a wafer as designed.

Interestingly, the resulting Te nanomesh patterns have a high micrometer-level resolution, which is determined by using a 1951 USAF resolution test target as a photomask (Fig. 5b, more SEM image shown in Supplementary Fig. 23). The resolution test (6 groups × 6 elements) provides a maximum resolution detectability of 228.1 line pairs per millimeter (lp/mm), corresponding to one of the best results in the microscale patterning of 1D nanomaterials. Based on the good patterning ability, we could use a halftone reprographic technique to generate continuous-tone imagery by altering the size and/or spacing of halftone dots (Fig. 5c and Supplementary Fig. 24). As demonstrated in Fig. 5d and Supplementary Fig. 25, different typical images, such as

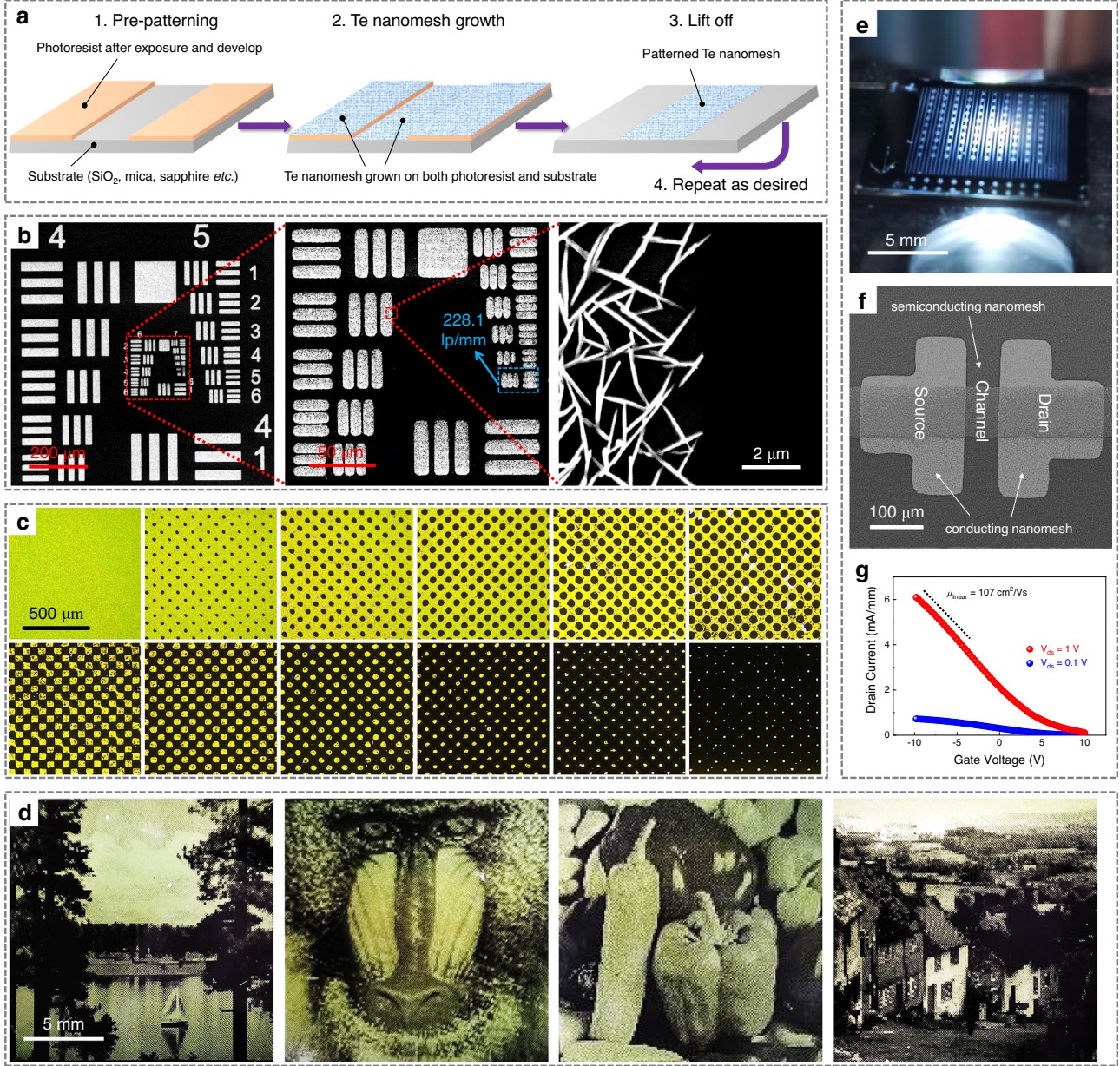

**Fig. 5 | Nanomesh patterning process. a** Schematic illustrations of the Te nanomesh patterning process. **b** SEM images of patterned nanomeshes in USAF 1951 resolution test. Red dashed lines indicate the enlarged position. The light blue arrow shows a maximum patterning resolution of 228.1 line pairs per millimeter (lp/ mm). **c** Optical images of halftone dots with different sizes and/or spacing that are made of patterned Te nanomeshes. **d** Optical images of Lake, Baboon, Peppers, and Goldhill images based on Te nanomeshes. **e** Optical image, **f** SEM image, and **g** transfer characteristic curves of the all-nanomesh TFT.

the Lake, Baboon, Peppers, and Goldhill, are made by patterned Te nanomesh through a halftone process. All the patterned images show high resolution and distinct edges, highlighting the robustness of large-area nanomesh patterning reported in this work. All these demonstrations of Te nanomesh patterning maintain both high resolution and cost-effectiveness, which could be extendable for other utilization.

Previously, patterning nanomaterials have mainly relied on the top-down "growth and etch" route, where it utilizes subtractive manufacturing to define and subsequently etch away unwanted parts. As high spatial resolution and repeated patterning are needed, it becomes time-consuming and difficult to handle, posing fabrication challenges[39]. Also, the surface contaminations that are introduced during etching would increase the chance for quality deterioration of as-grown nanomaterials. To avoid these issues, in this work, we thus designed and explored an additive manufacturing ("bottom-up") nanomesh patterning technique to maintain material quality and

possess µm-level resolution, being comparable to the masked growth of 2D materials (2 µm resolution)[39] and the direct lithography of inorganic nanocrystals (1 µm resolution)[40].

After that, we prototype an all-nanomesh TFT on the $SiO_2/p^{++}$ Si substrate (Fig. 5e, f, and Supplementary Fig. 26), where the nanomesh growth process and the standard photolithography process are executed twice to construct the all-nanomesh devices. Based on the tunable electrical properties of the Te nanomesh, the 3-h-grown semiconducting Te nanomeshes are used as the device channel, while the 5-h-grown metallic Te nanomeshes function as source/drain electrodes. It should be noted that prolonging the nanomesh growth duration (i.e., 5 h) could dramatically improve the welding status of NWs, featuring the increasing inter-NW connection, the enlarging soldering junctions, and the thicker NW bodies (as SEM images shown in Supplementary Fig. 27 and TEM images shown in Supplementary Fig. 6). All these morphological changes could work together to

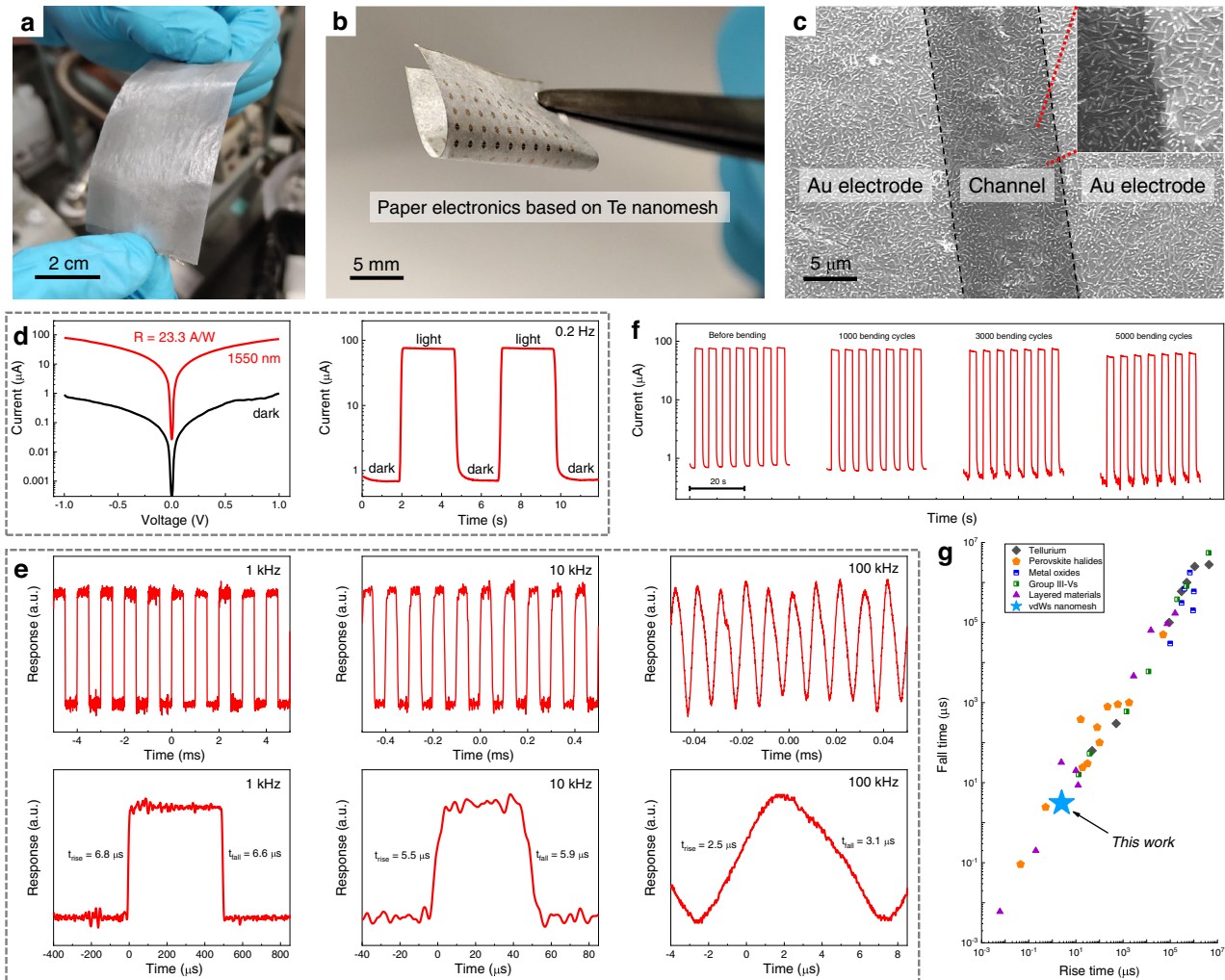

**Fig. 6 | Paper-based nanomesh infrared photodetectors. a** Photograph of wafer-scale Te nanomesh grown on paper substrates. **b** Photograph and **c** SEM image of Te nanomesh PD based on paper substrates. Red dashed lines indicate the enlarged position. **d** Photodetection performance of the paper-based Te nanomesh PD measured at 1550 nm irradiation with an incident power density of 4 mW/cm². **e** Photoresponse with different chopped frequencies from 1 kHz to 100 kHz. Underneath shows the corresponding high-resolution response with one switching on-off cycle. **f** Photoresponse with varying bending cycles during the bending test from 0 to 5000 times. **g** Comparison of the photoresponse speed of tellurium, perovskite halides, metal oxides, group III–Vs, layered materials, and Te nanomesh. The corresponding references could be found in Supplementary Table 4.

enhance the nanomesh conductivity. Indeed, after 5-h growth, the high conductivity of 266 S cm⁻¹ is obtained for the Te nanomesh, while also holds good optical transparency (>80%) in the visible region at the same time (Supplementary Fig. 28). Given the high conductivity and optical transmittance, together with the unique NW network morphology, the conducting Te nanomeshes could act well as the flexible p-type transparent conductors or device contact electrodes.

Evidently, the linear I-V relationships (from −1 V to 1 V) of the output curve of all-nanomesh TFTs indicate an ohmic-like electrical contact between the nanomesh channel and the nanomesh electrodes (Supplementary Fig. 29). From the transfer curves of all-nanomesh TFTs (Fig. 5g and Supplementary Fig. 29), all devices demonstrate the good p-type transistor performance with the peak hole mobility of $107 \pm 15$ cm²/Vs and the on-current of $6.0 \pm 0.5$ mA/mm. A slight TFT mobility reduction is observed after changing the contact electrodes from Au film to metallic Te nanomesh. However, the mobility is still higher than those of intrinsically flexible semiconductors (e.g., organic films or polycrystalline metal oxide nanofibers) that usually possess field-effect mobility of 1–10 cm²/Vs. The high mobility values from the all-nanomesh devices are related to the high-quality nanomesh fabrication and good compatibility with the nanomesh patterning process.

## Paper-based nanomesh infrared photodetectors

As discussed above, the goal of developing Te nanomesh fabrication is to realize low-cost and high-performance devices that can be implemented in large-area and cost-effective IoT applications. Paper is a kind of promising substrate for flexible and environmentally sustainable electronics[41]. Notably, the multi-substrate compatible growth process allows us to grow Te nanomesh on large-area paper substrates for device applications (Fig. 6a). To this end, by utilizing Te nanomesh as a light absorption component, we demonstrated the high-performance paper-based infrared photodetectors (PDs) (Fig. 6b). As depicted in the SEM image in Fig. 6c, the uniform well-connected Te nanomeshes are tightly attached to the paper surface, providing a navigable electrical channel between source/drain electrodes.

Room-temperature photodetector measurements are then carried out by using a 1550-nm infrared light source, which is an important wavelength for long-distance telecommunications[15,42]. Typically, crystalline Si or group III–V semiconductors dominate short-wave infrared (SWIR; from 1400 to 3000 nm) photodetector applications[42]. These PDs have great photodetector performance, but most involve complex growth/fabrication procedures and operate in a cooled environment[43]. On the opposite, our room-temperature operating nanomesh PDs

eliminate the cooling requirements, which can substantially reduce the corresponding detector size, cost, and power consumption on temperature control, thus extending their application scopes. When exposed to light illumination with an incident power density of 4 mW/cm$^2$, the photocurrent of the paper-based Te nanomesh device is significantly increased to ~78 μA, corresponding to a responsivity of 23.3 A/W and external quantum efficiency (EQE) of $2.9 \times 10^3$% (Fig. 6d). Such a responsivity has already outperformed the reported Te nanoflake infrared PDs[15] and the commercially available Si and InGaAs photodiodes[44].

The transient response is also critical for PDs and is highly dependent on the efficient transport/collection of photo-generated charge carriers[45–47]. The transient output signals of Te nanomesh PDs are measured at room temperature under modulated light illumination. As the photoresponse versus incident pulsed frequency depicted in Supplementary Fig. 30, the 3 dB cutoff frequency (i.e., the frequency at its 0.707 times peak photoresponse) of the paper-based Te nanomesh PD is estimated to be ~100 kHz. At the same time, the Te nanomesh devices exhibit good reliability and fast optical response in switching measurements with chopping frequencies ranging from 0.2 Hz, 1 kHz, 10 kHz, to 100 kHz (Fig. 6d, e; more data in Supplementary Figs. 31 and 32). In specific, at a chopping frequency of 100 kHz, the response times defined by the varying time for the photocurrent from 10 to 90% and from 90 to 10% are found to be 2.5 μs (rise time) and 3.1 μs (fall time), respectively (Fig. 6e). The μs-level response time is faster than most PDs reported in recent literature (as summarized in Fig. 6g and Supplementary Table 4)[48–50]. High material quality, high light absorption efficiency, and naturally terminated surfaces work together to contribute to the good PD performances of Te nanomeshes, particularly the rapid rise/fall time and strong photogating effect (Supplementary Fig. 21).

In addition to the good PD performance, our paper-based Te infrared PDs are also lightweight and mechanically robust for easy handling. To determine the mechanical durability of nanomesh PDs, the on-off switching behavior of photocurrent was evaluated with different bending cycles at a fixed 0.5-cm bending radius. No detectable photocurrent deterioration is observed through the entire bending test with bending cycles up to 5000 as presented in Fig. 6f. Also, bending test was performed on flexible Te nanomesh PDs based on thin mica substrates (Supplementary Fig. 33), where similar robust device performance was obtained with the bending test, which is mainly driven by the intrinsic superior mechanical properties of self-welded Te nanomeshes. Overall, the obtained PD performance metrics, including responsivity, response time, and device durability, are better than the reported Te PDs and comparable to those PDs based on state-of-the-art nanomaterials. Based on the technological importance of 1550-nm infrared photodetection, the combination of high-quality Te vdWs nanomeshes and low-cost paper substrates could be utilized in cost-efficient IoT applications, for instance, disposable (opto-)electronics or smart packaging[41].

### Nanomesh/layered material vdWs heterojunctions

Owing to the versatile fabrication process, Te nanomeshes could be heterogeneously integrated with other semiconductors to construct heterojunctions. As a kind of typical layered transitional metal chalcogenides semiconductors, tungsten disulfide (WS$_2$) is selected because of its good environmental stability and easy accessibility. In this work, monolayer WS$_2$, consisting of multi-domains (with an average domain size of ~10 μm)[51], is fabricated by a chemical vapor deposition method that is also compatible with the wafer-scale device technology[52]. After the Te nanomesh growth, the continuous Te nanomesh is witnessed on the top surface of the WS$_2$ layer from the detailed TEM study (Fig. 7a–c; more TEM images are shown in Supplementary Fig. 34). In the WS$_2$ regions, explicit lattice fringes are identified in the high-resolution TEM image in Fig. 7c, where the lattice

spacing is found to be 2.7 Å, corresponding to the {100} planes of WS$_2$ crystals. This observation reveals that the low-temperature Te nanomesh synthesis does not alter the crystalline properties of the underlying WS$_2$ monolayer, eliminating the possible adverse influence of the interfacial chemical reactivity or interdiffusion that would lead to device performance degradation in the subsequent studies.

To determine the relationship between the Te nanomesh and the underlying WS$_2$ layer, SAED study was carried out on the heterojunction region (Supplementary Fig. 35). Remarkably, there are three sets of diffraction spots found in the SAED pattern that are collected from the Te/WS$_2$ heterojunction region, in which two sets belong to the welded Te NWs and one set belongs to the underlying WS$_2$ layer. The SAED pattern suggests that the Te nanomesh vdWs epitaxially grow on the WS$_2$ layer, supporting the proposed growth mechanisms developed in this work related to multi-scale vdWs interactions. After that, EDS mappings are also performed to evaluate the 1D/2D heterostructure, where S is uniformly distributed throughout the entire heterostructure and Te is concentrated in the nanomesh regions (Fig. 7d and Supplementary Fig. 36). Similar results are also found in the Raman mapping study (Fig. 7f and Supplementary Fig. 37). All these material characterizations confirm the structural integrity of the 1D/2D mixed-dimensional vdWs heterostructure here.

After the successful construction of Te nanomesh/WS$_2$ heterostructures, their photoelectric performance is also investigated. After applying the 532-nm light irradiation with a power density of 1 mW/cm$^2$, the photocurrent of the Te nanomesh/WS$_2$ PD is significantly increased by 100,000 times (from 0.15 nA to 17 μA) when compared to the intrinsic WS$_2$ layer (Fig. 7g, h), together with a significant responsivity enhancement from 5 mA/W to 566.7 A/W. Impressively, the Te/WS$_2$ heterostructure devices exhibit good reliability and fast optical response in switching measurements with chopping frequencies ranging from 1 kHz, 10 kHz, to 100 kHz (Supplementary Fig. 38). In specific, at a chopping frequency of 100 kHz, the response times are found to be 3.0 μs (rise time) and 4.3 μs (fall time), respectively, being close to those of intrinsic Te nanomesh PDs (Supplementary Fig. 39). The responsivity and response time of the Te nanomesh/WS$_2$ monolayer mixed-dimensional heterostructure already outperforms most of the WS$_2$-based PDs reported until now. As the Te nanomesh has weak optical absorption efficiency in the region of 532 nm, the significantly improved photodetector performance can only be contributed by the band alignment at the Te nanomesh/WS$_2$ heterojunction.

In principle, the type-II band alignment in the Te nanomesh/WS$_2$ heterojunction could facilitate the separation of photo-generated electron–hole pairs, giving rise to the improved photoelectric performance of mixed-dimensional vdWs PDs (Supplementary Fig. 40)[53]. First, the interfacial charge transfer process after contact was studied by X-ray photoelectron spectroscopy (XPS), as judged by the blue shifts (~0.2 eV) of W 4$f$ peaks and S 2$p$ peaks in WS$_2$ after Te nanomesh deposition (Supplementary Fig. 41)[45]. This finding indicates the spontaneous electron transfer from WS$_2$ to Te, which results in the energy level bending and the formation of type-II band alignments. Also, the interfacial separation of photo-generated electron–hole pairs is supported by the photoluminescence (PL) study in Fig. 7e. The intrinsic WS$_2$ layer exhibits strong excitonic emission with a dominant PL emission peak at 620 nm. In contrast, the Te nanomesh/WS$_2$ heterostructure shows apparent PL quenching, indicating that the separation of excitons suppresses their recombination luminescence[54].

## Discussion

We report the vapor-phase growth of Te nanomeshes achievable at only 100 °C based on the ubiquitous vdWs interactions and show that the self-welded nanomeshes can be reliably grown on arbitrary surfaces. The multi-scale vdWs interactions, existing among Te atomic chains as well as between Te nanomeshes and substrates, govern the

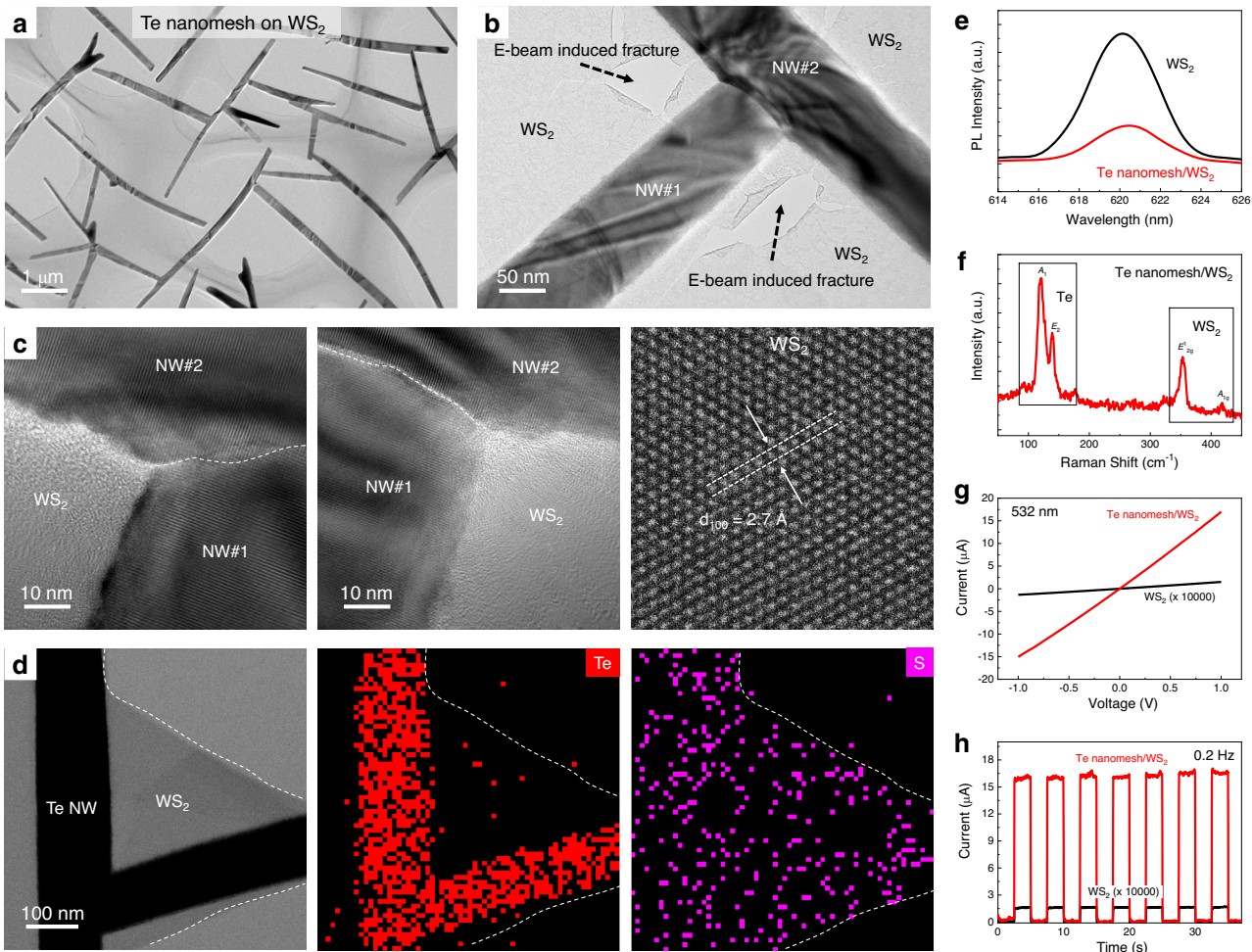

**Fig. 7 | Te nanomesh/layered material heterojunctions. a–c** TEM images of Te nanomesh/monolayer WS₂ mixed-dimensional heterostructures. White dashed lines in (**c**) represent the welding boundary between two Te NWs. **d** STEM image and energy-dispersive X-ray spectroscopy (EDS) mappings of Te nanomesh/monolayer WS₂ heterostructures. White dashed lines in (**d**) show the margin of lateral nanomesh growth and multiple-surface growth compatibility. monolayer WS₂. **e** PL and **f** Raman of Te nanomesh/monolayer WS₂ heterostructures. **g**, **h** Photodetector performance of Te nanomesh/monolayer WS₂ heterostructures measured at 532 nm irradiation with an incident power density of 1 mW/cm².

The rationally designed nanomeshes exhibit micrometer-level patterning capacity, high field-effect hole mobility, fast photoresponse in the optical communication region, and controllable electronic structure of the mixed-dimensional heterojunctions. All these findings can pave the way to the large-area assembly of functional vdWs nanomesh with potential applications not available by other means.

## Methods

### Substrate preparation

Prior to the nanomesh growth, SiO₂/p⁺⁺ Si, sapphire, and polyimide substrates were ultrasonically cleaned (in acetone, ethanol, and then deionized water, successively) and dried by nitrogen gas. Weighing papers and carbon fibers were used as purchased without any cleaning procedures. Monolayer WS₂ used in this work was grown on sapphire by using the conventional chemical vapor deposition[51]. Few-layer BN was exfoliated from bulk crystal using 3M tape and then transferred onto SiO₂/p⁺⁺ Si for further utilization. Mica sheets with layered structures were mechanically cleaved into transparent thin flakes before use. To prepare PDMS substrates, silicone elastomer (Sylgard 184) was mixed with the curing agent in a weight ratio of 10:1 and then poured into the module to form the 600-μm-thick layers. After that, silicone elastomer films were cured at 70 °C for 4 h and then 100 °C for 30 min.

### Te nanomesh synthesis

The vapor-phase growth of Te nanomeshes was carried out in a two-zone vapor transport system. For the typical growth process, a quartz tube with 1-in. inner diameter was utilized in the vapor transport system, while for the large-area growth, a quartz tube with 2-in. inner diameter was used instead. Then, 1 g of Te powders (Aldrich, pieces, 99.999%) was placed inside the high-temperature heating zone (450–500 °C), while growth substrates were positioned inside the low-temperature heating zone (100 °C). Before heating, the system was flushed with argon gas (200 sccm) for 30 min to remove any possible residual oxygen. To start the nanomesh growth, the Te source powder was vaporized at the setting temperature and carried to the low-temperature zone by argon gas (50 sccm). The growth pressure was maintained at atmospheric pressure during the entire growth process. In this work, the NW density (μm⁻²) was calculated from the number of Te NWs making up nanomeshes divided by the nanomesh coverage area. The repeated extractions of NW densities over different positions of one sample were performed to improve the data accuracy.

### Material characterization

The sample morphologies were examined by scanning electron microscopy (field-emission SEM, FEI Quanta 450). To prepare samples for transmission electron microscopy study (TEM, Philips CM-20; HRTEM, JEOL 2100F; HAADF-STEM, JEM-ARM300F2), the

Te nanomeshes and Te nanomesh/WS$_2$ heterostructures were transferred onto the Cu grids using a conventional polymer-based wet-transfer method. An energy-dispersive X-ray (EDS) detector attached to JEOL 2100F was utilized to perform elemental mappings. A Philips powder X-ray diffractometer (XRD) was used to determine the crystal structures. The PL spectra were acquired by F-4600 FL spectrophotometer with an excitation wavelength of 532 nm. To study the chemical states of samples, X-ray photoelectron spectroscopy (XPS, Thermo ESCALAB 250Xi system) was employed. The Raman study was done by an NTEGRA Spectra system (NT-MDT). The contact angles of water drop on various substrates were performed using a DataPhysics OCA 15EC Contact Angle Tester. Cross-sectional Te/SiO$_2$ samples were prepared by focused ion beam (FIB, FEI Scios Dual Beam system), and then STEM characterizations were conducted on atomic-resolution JEM-ARM300F2 operating at 300 KV. In situ friction property tests were performed with a Hysitron PI85 SEM PicoIndenter equipped on SEM (FEI Quanta 450 FE-SEM). Displacement control mode with a constant loading velocity of 20 nm/s was used during loading. The load-displacement data was recorded from the nanoindenter; meanwhile, the sample displacement was recorded via in situ SEM imaging.

### Nanomesh patterning process

Commercial photoresist solution was spin-coated onto the SiO$_2$/p$^{++}$ Si substrate at 4000 rpm and then thermally cured. Negative photoresist patterns were generated by making the areas exposed to light and removing materials from the exposed regions with a developer solution. After that, Te nanomeshes were grown on both the pre-patterned photoresist and the uncovered substrates at the same time, with a growth time of 4 h. After nanomesh growth, the patterned photoresist and the nanomeshes on photoresist were removed through a lift-off process, while only the nanomeshes grown on the uncovered substrates remained. The remaining patterned Te nanomeshes are capable of projecting microscale images as designed. A USAF (United States Air Force) 1951 resolution test target was used as a photomask to determine the patterning resolution. The test target used in this work has six groups (from +2 to +7), where each group has six elements, contributing to a maximum resolution detectability of 228.1 line pairs per millimeter (lp/mm). For image patterning of Lake, Baboon, Peppers, and Goldhill images, halftone patterns made on polyester-based films were used as photomasks.

### Device fabrication

Thin-film transistors (TFTs) were fabricated on the SiO$_2$/p$^{++}$ Si substrates, in which SiO$_2$ and p$^{++}$ Si act as the dielectric and gate layers, respectively. A growth time of 3 h was set to manufacture the device channel. After the nanomesh growth, 5/80-nm-thick Cr/Au films are deposited by thermal evaporation with a shadow mask to define the source and drain regions. To fabricate all-nanomesh TFT on SiO$_2$ dielectric layer, the nanomesh growth process and the photolithography process were executed twice. The 3-h-grown Te nanomeshes are used as the device channel, while the 5-h-grown conducting Te nanomeshes function as source/drain electrodes.

### Electrical characterization

Agilent 4155C semiconductor analyzer was employed to measure the TFT performance in a dark environment. The gate voltage sweeps from 10 V to −10 V, while the source-drain voltage is biased at −0.1 V or −1 V. To better show the current carry capacity of Te nanomeshes, the drain current density was calculated from the maximum drain current divided by the channel width. The field-effect mobility in linear regime ($\mu_{lin}$) was calculated with the equation of $\mu_{lin} = (L \times g_m)/(W \times V_{ds} \times C)$, where $L$ is the channel length, $W$ is the channel width, $g_m$ is the transconductance, $V_{ds}$ is the source-drain voltage, and $C$ is

the areal capacitance of 69 nf cm$^{-2}$ for 50-nm-thick gate dielectric of SiO$_2$. For photodetector measurements of the intrinsic Te nanomeshes and the Te nanomesh/WS$_2$ heterostructures, 1550 nm and 532 nm laser diodes are used as light sources, respectively. The incident light powers were measured by a power meter (PM400, Thorlabs). The responsivity ($R$) is defined as $R = I_{light}/(P \times A)$, where $I_{light}$, $P$, and $A$ are the photocurrent, incident power density, and effective irradiated area, respectively. The external quantum efficiency (EQE) was calculated with the equation of EQE = $(I_{light}/e)/(P \times A/h\nu)$, where e and h$\nu$ are the electronic charge and energy of an incident photon, respectively.

## Data availability

Relevant data supporting the key findings of this study are available within the article, the Supplementary Information file, and the Source Data file. All raw data generated during the current study are available from the corresponding authors upon request. Source data are provided with this paper.

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

## Acknowledgements

This work is supported by a fellowship award (CityU RFS2021-1S04) (J.C.H.) and the Theme-based Research (T42-103/16-N) (J.C.H.) of the Research Grants Council of Hong Kong SAR, China, the Foshan Innovative and Entrepreneurial Research Team Program, China (No. 2018IT100031) (J.C.H.), Beijing Municipal Natural Science Foundation (Grant No. 4204090) (L.F.S.), and City University of Hong Kong (project no. 7005650) (J.C.H.). We acknowledge the support of TEM facilities of TRACE Center, City University of Hong Kong.

## Author contributions

Y.M. and J.C.H. conceived and initiated the project. Y.M. carried out the sample growth, material characterization, device fabrication, and data analysis. X.C.L. and Y.L. conducted the in situ mechanical property characterization and analysis. X.C.L., W.P.L., and F.R.C. conducted the STEM characterization and analysis. X.L.K. and W.J.W. synthesized the layered substrates. Z.X.L. and Q.Q. contributed to the substrate preparation. W.W. performed the TEM characterization. Z.X.L. performed the optical transmittance measurement. X.M.B., S.P.Y., D.C., C.F.Y., L.F.S., C.T.L., and C.Y.W. contributed to data analysis. P.S.X., D.J.L., C.Y.L., and F.W. helped with electrical measurement. Y.M., X.C.L., F.R.C., F.W., C.Y.W., and J.C.H. wrote and revised the manuscript. All authors discussed the results and commented on the manuscript.

## Competing interests

The authors declare no competing interests.
