## [Peer Review File · Nature Communications]

Van der Waals nanomesh electronics on arbitrary surfacesEditorial Note: This manuscript has been previously reviewed at another journal that is not operating a transparent peer review scheme. This document only contains reviewer comments and rebuttal letters for versions considered at Nature Communications.

REVIEWER COMMENTS

Reviewer #1 (Remarks to the Author):

I recommend acceptance of this manuscript after the authors fully address the following comments.

- 1. The TLM method is not valid for such a long channel length. The contact resistance extraction would be inaccurate, given the long channel length.**
- 2. Why does the nanomesh configuration improve the mobility of the device?**
- 3. The heterojunction band diagram in S40 needs more careful examination. For example, what is the energy level for the CBM and VBM of the WS₂ and Te? Where is the fermi level of these materials? It looks like both materials are degenerated.**
- 4. Please show the data of 5d in a semi-log scale for the photo-ON/OFF ratio and zero-point bias.**

Response to Reviewers' Comments
on Nature Communications Manuscript NCOMMS-22-48872-T

We appreciate the referees for considering our manuscript and providing valuable comments. Based on the reviewer's further inputs, modifications, highlighted in red, have been made to the revised manuscript. Details can be found in the point-by-point response shown below.

Reviewers' Comments:

Reviewer #1 (Remarks to the Author):

I recommend acceptance of this manuscript after the authors fully address the following comments.

Reply:

We thank the referee for his/her positive comments on this manuscript. All the comments are important for us to improve the quality of the manuscript.

1. The TLM method is not valid for such a long channel length. The contact resistance extraction would be inaccurate, given the long channel length.

Reply:

We thank the important comment. Just as the reviewer proposed, the contact resistance extraction using TLM method for long-channel devices would possibly be inaccurate because the devices' performances are dominated or significantly influenced by channel resistance. We agree with this argument. Ideally, when contact resistance dominates the total resistance or is comparable to the channel resistance, the extraction of contact resistance would be more reliable. Based on this argument, it seems like the short-channel TLM method is preferred to be conducted for contact resistance extraction.

However, due to the considerable NW lengths (~ several μm) and randomly distributed grain boundaries (i.e., Schottky barriers) in Te nanomesh, the short-channel TLM method would generate deviation of contact resistance in statistics. For short-channel devices, the channel resistance and its variation need to be sufficiently small; otherwise, a small variation in channel resistance could lead to substantial errors in contact resistance extraction. Thus, we did not conduct the TLM method on short-channel devices with nanomesh channel lengths less than 10 μm .

Overall, based on the concerns about the TLM study on both long-channel and short-channel devices, the related TLM experiments (Supplementary Fig. S19g and S19h) and discussions (in Supplementary Text 4) have been removed from page S-20 of the revised Supplementary Information. This modification would not influence the main content and conclusion of the manuscript.

2. Why does the nanomesh configuration improve the mobility of the device?

Reply:

We appreciate the referee for pointing out this question. In this work, the fabricated Te nanomesh devices show improved field-effect hole mobility in some cases with different material systems, channel structures, etc. The reasons for the altering mobility when compared

with different types of channel counterparts are discussed from the viewpoints of Te material property, self-welding process, and nanomesh geometry.

First, comparing to scalable p-type semiconducting thin films, the obtained hole mobility value ($\sim 145 \text{ cm}^2/\text{Vs}$) of Te nanomesh in this work outperforms most of the p-type polycrystalline thin films, including metal oxides, metal halides, perovskites, organic materials, and evaporated Te-based films (detailed performance parameters can be found in Supplementary Table S3). The intrinsically high hole mobility of Te materials up to thousands cm^2/Vs , few crystal defects of individual Te NW among the nanomesh, and relatively few grain boundaries in the channel are the main reasons for the higher hole mobility of the nanomesh as compared to the conventional p-type polycrystalline semiconducting thin films. In this work, all the morphological and crystallographic features of the Te nanomesh are evaluated in detail and confirmed by SEM, TEM, and HAADF-STEM studies. These results show that the material properties of Te nanomesh could be well controlled, enabling effective modulation and improving the device performance, including the field-effect hole mobility.

Second, the obtained mobility is higher than those of NW networks or metal oxide nanofiber networks that usually possess field-effect mobility values of $1\sim 10 \text{ cm}^2/\text{Vs}$. Our results and the reported literature show that the self-welding process of NW networks or nanomesh is essential for promoting their performance in electrical devices. Compared to the NW networks formed by solution-phase deposition schemes that always give the weak physical inter-NW connection, the self-welded nanomesh with well-connected network morphology is expected to lower the contact barrier and reduce inter-NW junction resistance, making them a potentially high-performance device channel. To better show the nanomesh growth control, self-welding process, and corresponding device performance optimization of Te nanomesh, the related experiments were carried out as a function of different growth durations (Supplementary Fig. S7) and deposition temperatures (Supplementary Fig. S21). With the tunable diameter, length, and welding status of the Te nanomesh, we could obtain a platform to achieve hole mobility of $\sim 145 \text{ cm}^2/\text{Vs}$ in a controllable and reliable way.

Third, we would like to note that the obtained hole mobility values of Te nanomesh in this work are still significantly lower than some single-crystal Te nanostructures, like solution-grown 2D Te layers of $700 \text{ cm}^2/\text{Vs}$ [Nat. Electron. 2018, 1, 228], 1D Te NW encapsulated in a nanotube of $600 \text{ cm}^2/\text{Vs}$ [Nat. Electron. 2020, 3, 141], and single Te NW up to $1390 \text{ cm}^2/\text{Vs}$ [Adv. Funct. Mater. 2021, 31, 2006278], which is reasonable when considering the wafer-scale coverage and inter-NW connections among nanomesh. Also, the theoretical works proposed that the Te materials have potentially high hole mobility up to $\sim 10^5 \text{ cm}^2/\text{Vs}$ and small carrier effective mass, especially along the y direction [Sci. Bull. 2018, 63, 159]. Thus, there is still ample spacing to further improve the mobility of Te nanomesh with various strategies in the future.

In order to better describe the relationships among mobility, materials, and nanomesh configuration as well as to show the mobility level of Te nanomesh when compared with other material systems, the above discussion has been added to Supplementary Text 7 on page S-23 of the revised Supplementary Information.

3. The heterojunction band diagram in S40 needs more careful examination. For example, what is the energy level for the CBM and VBM of the WS_2 and Te? Where is the fermi level of these

materials? It looks like both materials are degenerated.

Reply:

We appreciate the referee for the valuable comment. To clearly show the energy level diagram of type-II band alignments in the Te nanomesh/WS₂ heterostructure, the energy level positions verified by ultraviolet photoelectron spectroscopy (UPS) were added to Supplementary Fig. S40 on page S-43 of the revised Supplementary Information (also shown below).

Supplementary Fig. S40 UPS analysis for (a) valence-band regions and (b) secondary electron cutoff regions of Te nanomesh and WS₂, respectively. (c) Energy level diagram of type-II band alignments in the Te nanomesh/WS₂ heterostructure before contact, after contact, and under light irradiation.

In principle, the type-II band alignment in the Te nanomesh/WS₂ heterojunction could facilitate the separation of photo-generated electron-hole pairs, giving rise to the improved photoelectric performance of mixed-dimensional vdWs PDs (Supplementary Fig. S40) [Adv. Funct. Mater., 32, 2203003, 2022]. First, the interfacial charge transfer process after contact was studied by X-ray photoelectron spectroscopy (XPS), as judged by the blue shifts (~ 0.2 eV) of W 4f peaks and S 2p peaks in WS₂ after Te nanomesh deposition (Supplementary Fig. S41) [ACS Nano, 14, 12749, 2020]. This finding indicates the spontaneous electron transfer from WS₂ to Te, which results in the energy level bending and the formation of the type-II band alignment. Also, the interfacial separation of photo-generated electron-hole pairs is supported by the photoluminescence (PL) study in Fig. 6e. The intrinsic WS₂ layer exhibits strong excitonic emission with a dominant PL emission peak at 620 nm. In contrast, the Te nanomesh/WS₂ heterostructure shows apparent PL quenching, indicating that the separation of excitons suppresses their recombination luminescence [ACS Nano 9, 555, 2015]. The above discussion has been modified on page 17 of the revised manuscript.

4. Please show the data of 5d in a semi-log scale for the photo-ON/OFF ratio and zero-point

bias.

Reply:

According to the referee's suggestions, the data of Figure 5d was revised into a semi-log scale on page 13 of the revised manuscript (also shown below). A photocurrent on/off ratio of $\sim 10^2$ is obtained when the paper-based Te nanomesh PD is irradiated with an incident power density of 4 mW/cm^2 .

Figure 5. (d) Photodetection performance of the paper-based Te nanomesh PD measured at 1550 nm irradiation with an incident power density of 4 mW/cm^2 .

List of other changes

- On page 1, "configurations" is changed to "**configuration**" and "composing" is revised to "**composed**".
- On page 2, "strategy" is changed to "**strategies**" and "on" is revised to "**of**".
- On page 4, "assembly" is changed to "**assemble**".
- On page 5, "instances" is changed to "**instance**".
- On page 6, "stop" is changed to "**stops**" and "determining" is revised to "**determined**".
- On page 9, "direct" is changed to "**the direct**".
- On page 10, "a positive" is changed to "**positive**".
- On page 12, "microscale" is changed to "**the microscale**".
- On page 15, "bending" is changed to "**the bending**".
- On page 18, "remained" is changed to "**remaining**".
- On page 19, "are" is changed to "**is**".
- Since new references (Ref 53 and Ref 54) have been added to the revised manuscript, the sequence of the references is re-arranged.
- Subtitles were revised in the manuscript according to the formatting requirements.
- Since a new figure (Supplementary Fig. S40) has been added to the revised Supplementary Information, the sequence of the figures is re-arranged.

-
- Since a new supplementary text (Supplementary Text 7) has been added to the revised Supplementary Information, the sequence of the supplementary text is re-arranged.

REVIEWERS' COMMENTS

Reviewer #1 (Remarks to the Author):

The authors have properly addressed my comments. I recommend acceptance of this manuscript for publication in Nature Communications.